# Spiking Transformer with Experts Mixture

**Zhaokun Zhou**[1,2]*, **Yijie Lu**[1]*, **Yanhao Jia**[3,7], **Kaiwei Che**[1,2], **Jun Niu**[1], **Liwei Huang**[4,2], **Xinyu Shi**[5,4], **Yuesheng Zhu**[1], **Guoqi Li**[6,2], **Zhaofei Yu**[5,4]†, **Li Yuan**[1,2]†

[1]School of Electronic and Computer Engineering, Shenzhen Graduate School, Peking University
[2]Peng Cheng Laboratory
[3] College of Computing and Data Science, Nanyang Technological University
[4]School of Computer Science, Peking University
[5]Institute for Artificial Intelligence, Peking University
[6]Institute of Automation, Chinese Academy of Sciences
[7]Deep NeuroCognition Lab, I2R and CFAR, Agency for Science, Technology and Research

## Abstract

Spiking Neural Networks (SNNs) provide a sparse spike-driven mechanism which is believed to be critical for energy-efficient deep learning. Mixture-of-Experts (MoE), on the other side, aligns with the brain mechanism of distributed and sparse processing, resulting in an efficient way of enhancing model capacity and conditional computation. In this work, we consider how to incorporate SNNs' spike-driven and MoE's conditional computation into a unified framework. However, MoE uses softmax to get the dense conditional weights for each expert and TopK to hard-sparsify the network, which does not fit the properties of SNNs. To address this issue, we reformulate MoE in SNNs and introduce the Spiking Experts Mixture Mechanism (SEMM) from the perspective of sparse spiking activation. Both the experts and the router output spiking sequences, and their element-wise operation makes SEMM computation spike-driven and dynamic sparse-conditional. By developing SEMM into Spiking Transformer, the Experts Mixture Spiking Attention (EMSA) and the Experts Mixture Spiking Perceptron (EMSP) are proposed, which performs routing allocation for head-wise and channel-wise spiking experts, respectively. Experiments show that SEMM realizes sparse conditional computation and obtains a stable improvement on neuromorphic and static datasets with approximate computational overhead based on the Spiking Transformer baselines.

## 1 Introduction

The spiking neural networks (SNNs) are regarded as the third generation of neural networks [1], distinguished by biological plausibility [2], spike-driven characteristic, and low power consumption. SNNs emulate the dynamics of biological neurons at a microscopic level, utilizing asynchronous binary spikes for information transmission. The membrane potential of spiking neurons in SNNs is only updated upon the arrival of spikes, avoiding calculations of zero values. The inherent features make SNNs promising candidates for low-energy consumption on neuromorphic hardware, such as TrueNorth [3] and Loihi [4]. There are lots of architectures in SNNs include Spiking Recurrent Neural Networks [5], ResNet-like SNNs [6–9], Spiking Graph Neural Networks [10], and Spiking Transformers [11–13]. Spiking Transformers stands at the forefront. Spikformer [11] introduces Spiking Self-Attention (SSA). The Spike-Driven Transformer [12] introduces Spike-driven Self-

---

*Equal
†Corresponding author

Attention. Other works explore the Spiking Transformer in terms of structural improvements [14–17], training methods [18], and different tasks [19], respectively.

Mixture-of-Experts (MoE) [20, 21] is known for allowing each expert to learn specific tasks or features, showing better performance, conditional computing and dynamic adaptability, which are crucial features in the brain mechanism [22, 23]. In this work, we are committed to exploring the effective integration of MoE and Spiking Transformer. As shown in Fig.1(a), MoE introduces a large number of parameters based on the original Transformer, and the conditional computation is achieved by calculating the routing probability of each token on each expert through the softmax function. Selecting Top-K experts based on the routing probability, MoE achieves hard sparsification. However, SNN calculations need to avoid multiplication and cannot use complex softmax functions. The features in SNNs are dynamically sparse and do not require additional TopK. All the parameters of the expert must be loaded, which aggravates the difficulty of neuromorphic chip deployment. These factors make porting MoE to SNNs non-trivial. To tackle this problem, we develop the Spiking Experts Mixture Mechanism (SEMM), as shown in Fig.1(b), a universal SNN-MoE paradigm with the following three main features. 1) The SEMM is spike-driven. The outputs of the expert and the router are spike sequences, and the element-wise operation between them conforms to the SNN characteristics, i.e., avoiding multiplication. 2) SEMM leverages the sparse spiking activation of SNNs to achieve dynamic conditional computation of MoEs, which is more flexible than the fixed hard sparsification of Artificial Neural Network MoE (ANN-MoE). 3) With reasonable parameter count settings, SEMM enables Spiking Transformers to achieve stable performance gains with negligible overhead.

Based on SEMM, we modify the Spiking Self-attention (SSA) and Multi-layer Perceptron (MLP) of Spiking Transformers to obtain the Experts Mixture Spiking Attention (EMSA) and the Experts Mixture Spiking Perceptron (EMSP). EMSA treats each head of SSA as an expert, computes respective attention, and employs a temporal-aware router to integrate attention. As for Multi-layer Perceptron (MLP), it is common to substitute the entire MLP with MoE [24], which leads to better performance but a significant increase in the overall parameter number. EMSP implements a channel-wise MoE within MLP to

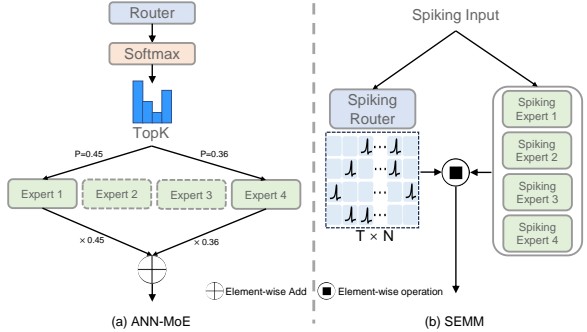

Figure 1: ANN-MoE and Spiking Experts Mixture Mechanism (SEMM). $N$ denotes the length of image patches.

overcome the shortcoming. EMSA and EMSP can be inserted directly and seamlessly into existing Spiking Transformer variants [11, 14, 12]. Our work contributes in three main aspects:

- We introduce the Spiking Experts Mixture Mechanism (SEMM), a universal SNN-MoE paradigm. SEMM is spike-driven, capable of efficient dynamic sparse conditional computation.

- Based on SEMM, we develop the Experts Mixture Spiking Attention (EMSA), whose information from all head-wise experts is selectively integrated through a temporal-aware router. We restructure MLP by a channel-level spiking-sparse SEMM, named the Experts Mixture Spiking Perceptron (EMSP). They can seamlessly replace self-attention and MLP in Spiking Transformers.

- Extensive experiments demonstrate the stable performance improvement of SEMM on both static and neuromorphic datasets. Notably, Spike-driven Transformer-8-512 with SEMM achieves a remarkable 76.62% accuracy on ImageNet with 4 time steps, surpassing the baseline (74.57%).

## 2   Related Work

**Deep Spiking Neural Networks and Spiking Transformers.** Spatio-temporal backpropagation(STBP) [25] directly trains SNNs by performing backpropagation on both spatial and temporal domains. Temporal backpropagation [26] computes the gradients of the timings of existing spikes for the membrane potential at the spike timing. Treshold-dependent batch normalization (tdBN) [8] is used to extend the network depth. SEW-ResNet [7] proposed the spiking element-wise residual for SNNs. Spikformer [11] firstly converts all the components of Vision Transformer (ViT) into

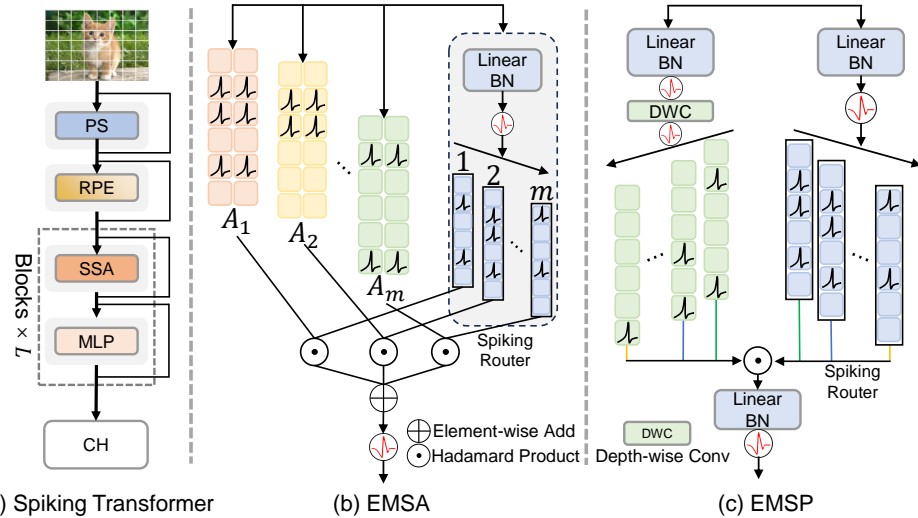

(a) Spiking Transformer     (b) EMSA     (c) EMSP

Figure 2: (a) The overview of Spiking Transformer. (b) The Experts Mixture Spiking Attention (EMSA). (c) The Experts Mixture Spiking Perceptron (EMSP). EMSA and EMSP can directly replace SSA and MLP in (a).

spike-form and pioneers the field of SNN-Transformer. Spike-driven Transformer [12] goes further by introducing linear spike-driven self-attention. Spikingformer [14] proposes a hardware-friendly spike-driven residual learning architecture. Besides, Masked Spiking Transformer [27] combines SNNs and Transformers from the perspective of the ANN-to-SNN conversion. However, the SNN Transformer architecture that can bring SNNs' superiority into full play is still ongoing research.

**Mixture-of-Experts.** The Mixture-of-Experts (MoE) [28, 29] combines the predictions of multiple specialized experts, which is effective in handling high-dimensional data and complex problems. Researchers explore diverse gating mechanisms [30, 31], optimizing expert allocation strategies [32, 33], and enhancing the scalability of MoE models [34, 35]. Sparsely-Gated Mixture-of-Experts [30] adopted MoE into architectures such as the Long Short-Term Memory (LSTM) [36], showcasing effectiveness in Language Modeling. The Transformer also benefits from MoE with the substitution of the Multi-layer Perceptron (MLP) [24, 37]. Switch Transformer [35] has scaled the models with trillions of parameters. Currently, there is no existing work on MoE with Spiking Transformers.

## 3 Methodology

### 3.1 Preliminaries and Overall Architecure

**Spiking neuron** is the basic unit of SNNs. For the dynamics of the Leaky Integrate-and-Fire (LIF) neuron used in this work, the $t-th$-time-step membrane potential $U[t]$ is equal to the sum of the state potential $H[t-1]$ at the previous time step and the input $X[t]$. When membrane potential exceeds the threshold $u_{th}$, the neuron will fire a spike, otherwise, it remains inactive. Consequently, the output $S[t]$ only contains binary values, either 1 or 0. $\mathrm{Hea}(\cdot)$ is a Heaviside function that satisfies $\mathrm{Hea}(x) = 1$ when $x \geq 0$, otherwise $\mathrm{Hea}(x) = 0$. $H[t]$ is the temporal state output, and $V_{reset}$ denotes the reset potential after a spiking event. $\beta < 1$ determines the rate of decay. If the neuron remains inactive, the potential $U[t]$ decays towards $H[t]$ over time. LIF can be described as:

$$U[t] = H[t-1] + X[t], \tag{1}$$

$$S[t] = \mathrm{Hea}\left(U[t] - u_{th}\right), \tag{2}$$

$$H[t] = V_{reset}S[t] + (\beta U[t])(1 - S[t]). \tag{3}$$

**Spiking Transformer**[11, 14, 12] baselines contain Patch Splitting (PS), Relative Position Embedding (RPE), Spiking Self Attention (SSA) (e.g., Spiking Self-Attention [11] and Spike-driven Self-Attention[12]), MLP and linear classification head, as shown in Fig. 2(a). Given a 2D image sequence $\mathbf{I} \in \mathbb{R}^{T \times C \times H \times W}$, where $T$, $C$, $H$, and $W$ denote time-step, channel, height and width, the PS splits it into a sequence of $N$ flattened spike patches $\mathbf{x}$ with $D$ dimensional channel. A Convolution-BatchNorm-LIF block generates Relative Position Embedding (RPE):

$$\mathbf{X}_0 = \mathrm{PS}\,(\mathbf{I}) + \mathrm{RPE}, \tag{4}$$

$$\mathbf{X}_l^{'} = \mathrm{SSA}(\mathbf{X}_{l-1}) + \mathbf{X}_{l-1}, \tag{5}$$

$$\mathbf{X}_l = \mathrm{MLP}(\mathbf{X}_l^{'}) + \mathbf{X}_l^{'}, \tag{6}$$

$$\mathbf{Y} = \mathrm{CH}(\mathrm{GAP}(\mathbf{X}_L)), \tag{7}$$

where the $\mathbf{X}_0$ is fed into the $L$-blocks and each block consists of a SSA and a MLP. $\mathbf{X}_l^{'}, \mathbf{X}_l$ are spike sequences and $l = 1...L$ is layer. A Global Average-pooling (GAP) is utilized on the $\mathbf{X}_L$ and the linear Classification Head (CH) to output the prediction $\mathbf{Y}$. See Appendix. A for more details.

### 3.2 Spiking Experts Mixture Mechanism

Before exploring the adaptation of spiking MoE in SNNs, we first review MoE in ANNs. An ANN-MoE layer typically comprises a set of $m$ experts $\mathbf{E}_\mathrm{A} = \{\mathbf{E}_1, \mathbf{E}_2, ..., \mathbf{E}_m\}$, along with a router $\mathcal{R}_\mathrm{A}$ for selecting the corresponding experts. Given an input sequence $\mathbf{X}_\mathrm{A}$, the resulting output can be expressed as the sum of the Top-K selected experts from $m$ candidates using a router:

$$\mathrm{y} = \sum_{k=1}^{K} \mathcal{R}_k(\mathbf{X}_\mathrm{A}) \cdot \mathbf{E}_k(\mathbf{X}_\mathrm{A}), \tag{8}$$

$$\mathcal{R}(\mathbf{x}) = \mathrm{TopK}(\mathrm{softmax}(\mathbf{X}_\mathrm{A}\mathbf{W}_\mathcal{R}, K)), \tag{9}$$

$$\mathrm{TopK}(\boldsymbol{v}, K) = \left\{ \begin{array}{ll} \boldsymbol{v} & \text{if } \boldsymbol{v} \text{ is in the top } K \text{ elements} \\ 0 & \text{otherwise} \end{array} \right. \tag{10}$$

where $\mathbf{W}_\mathcal{R}$ is the router weight matrix. The $\mathrm{TopK}(\cdot, K)$ together with the $\mathrm{softmax}(\cdot)$ sets all elements of the routing vector to zero except the largest top $K$ values. The sparse conditional computing is obtained from the selecting of TopK and the different routing weights of the softmax, while $K$ is usually taken as $0.5 * m$. We argue that the ANN-MoE is not suitable for SNN for two main reasons. First, the float-point routing-expert and the softmax which involve exponentiation and division, do not adhere to the computation principles of SNNs. Second, SNN experts are inherently highly sparse, so the hard sparsification approach of additional TopK is unnecessary. An event-triggered spike-based sparse conditional computation on asynchronous neuromorphic chips is needed more than TopK. To bridge these gaps, we introduce a generalized representation of the Spiking Experts Mixture Mechanism (SEMM), which is as follows,

$$\mathrm{SEMM}(\mathbf{E}, \mathcal{R}, \mathrm{F}(\cdot)) = \mathrm{F}(\mathbf{E}, \mathcal{R}), \tag{11}$$

where $\mathbf{E} = \{\mathbf{E}_1, \mathbf{E}_2, ..., \mathbf{E}_m\}$ represents the spiking sequence of $m$ spiking experts in the Spiking Transformer, and $\mathcal{R} \in \{0, 1\}^{T \times N \times m}$ represents the spiking router for allocating computation. $\mathrm{F}(\cdot)$ denotes the element-wise form of the operation between the router and the expert output spiking sequence, i.e., Hadamard product and addition.

As a MoE mechanism specifically designed for SNNs, SEMM has the following three significant advantages. i) **Spike-driven.** The SEMM is spike-driven, which is important for SNNs. Due to the spike-driven computation mode of experts, i.e., Spiking Self-attention and Spiking-MLP, SEMM computations are triggered by sparse spikes of experts and require only synaptic manipulation. For example, the Hadamard product between the spiking signals $\mathcal{R}$ and $\mathbf{E}$ is equivalent to mask. ii) **Sparse-spiking conditional computation.** The SEMM subtly utilizes the sparse activation of the spiking routers for the conditional computation of MoE. SEMM has a variable sparsity when dealing with different data. Additionally, SEMM does not suffer from the load imbalance problem in ANN-MoE, i.e., TopK selects fixed number of experts. The sparse conditional computation is distributed to each expert. iii) **Efficient computation.** Unlike loading with multiple heavy expert modules of ANN-MoE, the SEMM has comparable parameters and operations to the previous SSA and MLP of Spiking Transformers. These are further discussed in Sec. 3.5. Without loss of generality, we use the mainstream architecture of Spiking Transformer for SEMM embedding in Sec. 3.3 and Sec. 3.4.

### 3.3 Experts Mixture Spiking Attention

We begin by reviewing the processing of Spiking Self-Attention (SSA). Different from vanilla ANN-Transformers [38], it discards the softmax normalization for the attention map. The SSA mechanism

can be described by the following equation:

$$\mathbf{Q} = \mathcal{SN}_{\mathbf{Q}}(\text{L-BN}_{\mathbf{Q}}(\mathbf{X})), \mathbf{K} = \mathcal{SN}_{\mathbf{K}}(\text{L-BN}_{\mathbf{K}}(\mathbf{X})), \mathbf{V} = \mathcal{SN}_{\mathbf{V}}(\text{L-BN}_{\mathbf{V}}(\mathbf{X})), \quad (12)$$

$$\mathbf{A} = \text{SSA}(\mathbf{Q}, \mathbf{K}, \mathbf{V}) = \begin{cases} \mathcal{SN}(\mathbf{Q}\mathbf{K}^{\text{T}}\mathbf{V} * s), \text{for Spiking Self-Attention} \\ \mathcal{SN}(\text{SUM}_{\text{c}}(\mathbf{Q} \odot \mathbf{K})) \odot \mathbf{V}, \text{for Spike-Driven Self-Attention} \end{cases} \quad (13)$$

where $\mathcal{SN}$ represents the spiking neuron and L-BN represents that the features pass sequentially through Linear and BatchNorm. $s$ is the scaling factor and $\mathbf{Q}, \mathbf{K}, \mathbf{V} \in \{0, 1\}^{T \times N \times D}$ are in spike-form. $\mathbf{A} \in \{0, 1\}^{T \times N \times D}$ is the spiking output of SSA. $\odot$ is the Hadamard product, and $\text{SUM}_{\text{c}}(\cdot)$ denotes the sum of each column. As shown in Fig 2(b), the Experts Mixture Spiking Attention (EMSA) based on SEMM and SSA is formulated as:

$$\mathbf{A_m} = \text{SSA}(\mathbf{Q_m}, \mathbf{K}, \mathbf{V}), \quad (14)$$

$$\mathbf{E} = \{\mathbf{A}_1, \mathbf{A}_2, ..., \mathbf{A}_m\}, \quad (15)$$

$$\mathcal{R} = \mathcal{SN}(\text{BN}(\mathbf{X}\mathbf{W}_{\mathcal{R}})) = \{\mathbf{r}_1, \mathbf{r}_2, ..., \mathbf{r}_m\}, \quad (16)$$

$$\text{SEMM}(\mathbf{E}, \mathcal{R}, \text{F}(\cdot)) = \sum_{i=1}^{m} \mathbf{r}_i * \mathbf{A}_i, \quad (17)$$

$$\text{EMSA} = \mathcal{SN}(\text{BN}(\text{SEMM}(\mathbf{E}, \mathcal{R}, \text{F}(\cdot)))\mathbf{W}_o), \quad (18)$$

where $\mathbf{A}_m \in \{0, 1\}^{T \times N \times d}$ is the output of each SSA experts, and $\mathbf{W}_{\mathcal{R}} \in \mathbb{R}^{D \times m}$ is the router weight matrix. $\mathcal{R} \in \{0, 1\}^{T \times N \times m}$ is the routing sequence. We set $d \leq D$ by default to avoid introducing too many parameters. The operation $\text{F}(\cdot)$ between expert-router is computed by masking the expert using router pair-by-pair and then summing them up. The output of EMSA is obtained by performing matrix multiplication between $\text{SEMM}(\mathbf{E}, \mathcal{R}, \text{F}(\cdot))$ and synaptic weight $\mathbf{W}_o \in \mathbb{R}^{d \times D}$, the same as the last layer of SSA block in Spiking Transformers.

### 3.4 Experts Mixture Spiking Perceptron

As illustrate in Fig. reffig:method, the Experts Mixture Spiking Perceptron (EMSP) launches the channel-wise SEMM within MLP. The architecture can be written as follows:

$$\mathbf{E} = \mathcal{SN}(\text{DWC}(\mathcal{SN}(\text{BN}(\mathbf{X}\mathbf{W}_1)))) \in \{0, 1\}^{T \times N \times m} \quad (19)$$

$$\mathcal{R} = \mathcal{SN}(\text{BN}(\mathbf{X}\mathbf{W}_{\mathcal{R}})) \in \{0, 1\}^{T \times N \times m}, \quad (20)$$

$$\text{SEMM}(\mathbf{E}, \mathcal{R}, \text{F}(\cdot)) = \mathbf{E} \odot \mathcal{R}, \quad (21)$$

$$\text{EMSP} = \mathcal{SN}(\text{BN}((\text{SEMM}(\mathbf{E}, \mathcal{R}, \text{F}(\cdot))\mathbf{W}_o)) \in \{0, 1\}^{T \times N \times D}, \quad (22)$$

where $\mathbf{W}_1, \mathbf{W}_{\mathcal{R}} \in \mathbb{R}^{D \times m}$ is the weight matrix of first layer in MLP and router, respectively. We integrate a $3 \times 3$ Depth Wise Convolution layer DWC to capture local features on each channel expert, which has fewer parameters and is computationally efficient compared to original convolution. $\mathbf{W}_o \in \mathbb{R}^{m \times D}$ is the wight matrix of output layer. The original Spiking Transformer's MLP would have a $D$ to $4 * D$ channel dimension increase after the first layer and the second layer would reduce the dimension back to $D$. In EMSP we set $m$ to $(8//3) * D$ to try to match the parameter number of the original MLP. EMSP integrates sparse routing of multiple channel-wise experts. It can also be viewed as the channel-wise gating, which is suitable for temporal information processing, allowing information to flow unimpeded through potentially many time steps [39]. The gate (router) branch and expert branch can be also regarded as incorporating the Spike Element-Wise (SEW) [7] residual block into EMSP. Channel-wise convolution and element-wise (Hadamard) products only introduce a minor increase in computational cost. By appropriately configuring the number and dimensions of expert networks, our EMSP achieves more efficient computation compared to the original spiking MLP. More details of the computational overhead are given in the next sub-section.

### 3.5 Characteristics of SEMM

We explain each of the three advantages of SEMM, i.e., Spike-driven, Sparse-spiking conditional computation and efficient computation.

**Spike-driven** has the following formal definition, meaning that the gathering of input current is initiated by sparse spikes released from presynaptic neurons:

**Definition 1.** *In SNN, the operation is spike-driven if the input currents satisfy the following form,*

$$I_i[t] = \sum_j w_{i,j} s_j[t] = \sum_{j, s_j[t] \neq 0} w_{i,j}, \tag{23}$$

*where $I_i[t]$ is the input current of the $i$-th postsynaptic neuron at time step $t$, $s_j[t] \in \{0, 1\}$ is the spike output of the $j$-th pre-synaptic neuron, $w_{i,j}$ is the weight of the synaptic connection from $j$ to $i$.*

For EMSA, the input current for the LIF in Eq. 18 at a specific time step $t$ is given by:

$$\mathbf{I}[t] = \text{SEMM}(\mathbf{E}[t], \mathcal{R}[t], \text{F}(\cdot))\mathbf{W}_o = \sum_{i=1}^m r_i[t] * \mathbf{A}_i[t]\mathbf{W}_o, \tag{24}$$

$$I_{p,q}[t] = \sum_{i=1}^m r_i[t] \sum_l a_{i,p,l}[t] w_{l,q} = \sum_{\substack{i,l \\ (r_i[t] \wedge a_{i,p,l}[t]) \neq 0}} w_{l,q}, \tag{25}$$

where $\mathbf{I}[t]$ has a dimension of $P \times Q$, with $p$ and $q$ represent the $p$-th row and $q$-th column, respectively. EMSA is essentially spike element-wise addition after masking operation on SSA, consistent with the characteristics of spike-driven. For EMSP, the operation between two spiking sequences $\mathbf{E}$ and $\mathcal{R}$ corresponds to the logical AND function [7], which also conforms to the spike driven,

$$\mathbf{I}[t] = \text{SEMM}(\mathbf{E}[t], \mathcal{R}[t], \text{F}(\cdot))\mathbf{W}_o = (\mathbf{E}[t] \odot \mathcal{R})[t]\mathbf{W}_o, \tag{26}$$

$$I_{p,q}[t] = \sum_l e_{p,l}[t] r_{p,l}[t] w_{l,q} = \sum_{\substack{l \\ (e_{p,l}[t] \wedge r_{p,l}[t]) \neq 0}} w_{l,q}. \tag{27}$$

**Sparse-spiking conditional computation** means using spiking routers to dynamically allocate the computation in temporal and spatial dimension. More analysis is detailed in experiments.

**Efficient computations** means that SEMM approximates SSA and MLP in terms of number of parameters and theoretical synaptic operations. Tab. 1 demonstrates the computation load for EMSA, EMSP versus SSA and MLP. In terms of parameter number, despite having a routing layer $mD$, EMSA has a smaller number of parameters than SSA when the number of experts is within a reasonable range ($\geq 2$). EMSP additionally introduces a $3 \times 3$ depthwise convolution, and the number is slighter higher than MLP. Due to the similarity of the actual spiking rates, EMSA is smaller than SSA on the $TND^2$ term, while the calculation of the additional introduced by SEMM is on the $TND$ term (much smaller than $TND^2$) and therefore can be ignored. The situation is similar on EMSP, with depth-wise convolution adding a slight computational

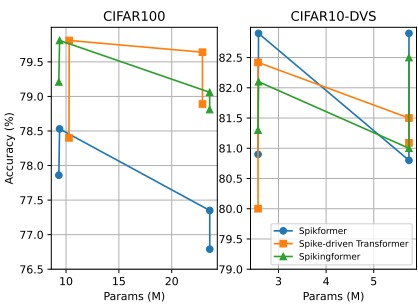

Figure 3: Comparison of parameters-accuracy for different MoEs. Methods of each line from left to right correspond to: 1. the baseline, 2. EMSP, 3. ANN-router with four heavy MLP experts, 4. Spiking-router with four heavy MLP experts, respectively. The last two use softmax and TopK ($K = 2$).

overhead. The design of EMSP is different from ANN-MoE, which selects multiple heavy MLPs as experts. We demonstrate it's validity in Fig. 3, i.e., it performs better while the number of parameters is much smaller than ANN-MoE.

## 4 Experiments

### 4.1 Sparse Conditional Computation Analysis

We analyze the average spiking rate (ASR) of routers for EMSA and EMSP on the ImageNet validation set, which is shown in Tab. 2. The SD-Transformer-8-512 is used for the analysis. The ASR of EMSA is around 0.5, which is comparable to the regular TopK setting of ANN-MoE. The ASR of EMSP is low and gradually decreases as the block deepens, compared to the fixed TopK hard sparse in ANN-MoE, SEMM fully reflects the advantage of SNN dynamic sparse conditional computation. To further verify the spiking router, we ablate it, i.e., cancel it in EMSA and EMSP, as

Table 1: Number of parameters and theoretical synaptic operations. $\overline{R}$, $\widehat{R}$, $\tilde{R}$ and $R$ denote the average spike firing rates (the proportion of non-zero elements in the spike matrix) in various spike matrices. $T$, $N$, and $D$ are the time step, sequence length, and channel dimension of the input features, respectively. $d$ is the channel dimension of $\mathbf{A}_m$ and $m$ is the number of experts. The details are provided in the Appendix. B.

|  | Param | OPs |
|---|---|---|
| SSA | $4D^2$ | $4\overline{R}TND^2 + 2\widehat{R}TN^2D$ |
| EMSA | $(1+1/m)D^2 + (2d+m)D$ | $(1+1/m)\overline{R}TND^2 + (2d+m)\tilde{R}TND + (D+md)RTN^2$ |
| MLP | $8D^2$ | $8\overline{R}TND^2$ |
| EMSP | $8D^2 + 24D$ | $8\overline{R}TND^2 + 24\tilde{R}TND$ |

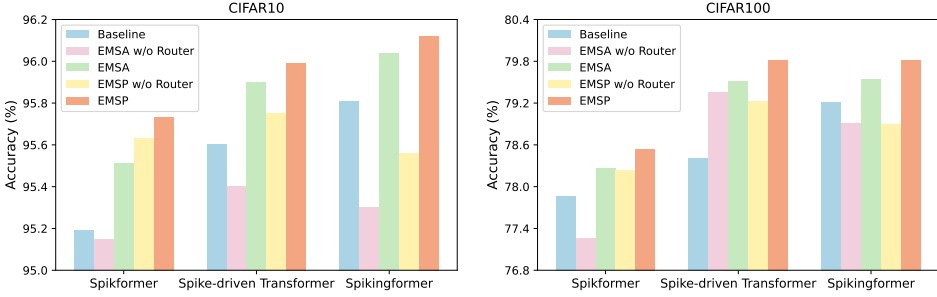

Figure 4: Ablation study on the EMSA and EMSP router.

shown in Fig. 4, the accuracy decreases significantly after canceling the router, and it is even lower than the baseline model on the Spikingformer. Directly analyzing the conditional computation of SEMM is challenging, we therefore use visualizations to illustrate this intuitively. As show in Fig. 5, for the same image, each expert's router assigns a different computational region, e.g., the third router filters the background of the expert's features, while the second router assigns the computation to the foreground target. The computation allocation of the spiking router to the irregular object "snake" can be seen that the router is highly dynamic and effective. See Appendix. D for more samples. In addition, as shown in Fig. 6, we also report the ASR of spatial-temporal locations of routers in different kinds of images. As can be seen by the difference in average firing rates, spiking router has a dynamic adjustment of ASR processing different kinds of images, further illustrating its data-dependent conditional computation property.

Table 2: The average spike rate of EMSA and EMSP router in 8 blocks testing on the ImageNet.

|  | Block0 | Block1 | Block2 | Block3 | Block4 | Block5 | Block6 | Block7 |
|---|---|---|---|---|---|---|---|---|
| EMSA | 0.64 | 0.68 | 0.49 | 0.52 | 0.47 | 0.43 | 0.51 | 0.50 |
| EMSP | 0.25 | 0.30 | 0.33 | 0.27 | 0.20 | 0.10 | 0.05 | 0.02 |

## 4.2 Results on various Datasets

We conduct experiments on static datasets, i.e., ImageNet [40] and CIFAR [41], and neuromorphic datasets, i.e., CIFAR10-DVS [42], DVS128 Gesture [43] to verify the effectiveness of SEMM. See Appendix. C for more details. **ImageNet** results are shown in Tab. 3 which mainly compares SEMM with the Spiking Transformer baselines. At slightly lower model parameter counts, SEMM is steadily superior to baselines. For instance, Spikformer-8-512 with SEMM is 2.55% higher than the baseline with $28.22M$ parameters, Spike-driven Transformer-8-384 with SEMM obtain 2.05% improvement. SEMM on Spikingformer presents similar findings.

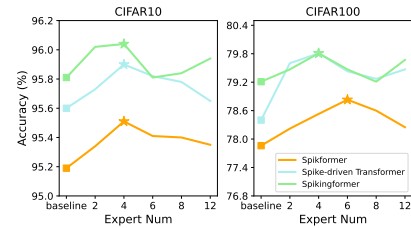

Figure 8: Accuracy with different experts number.

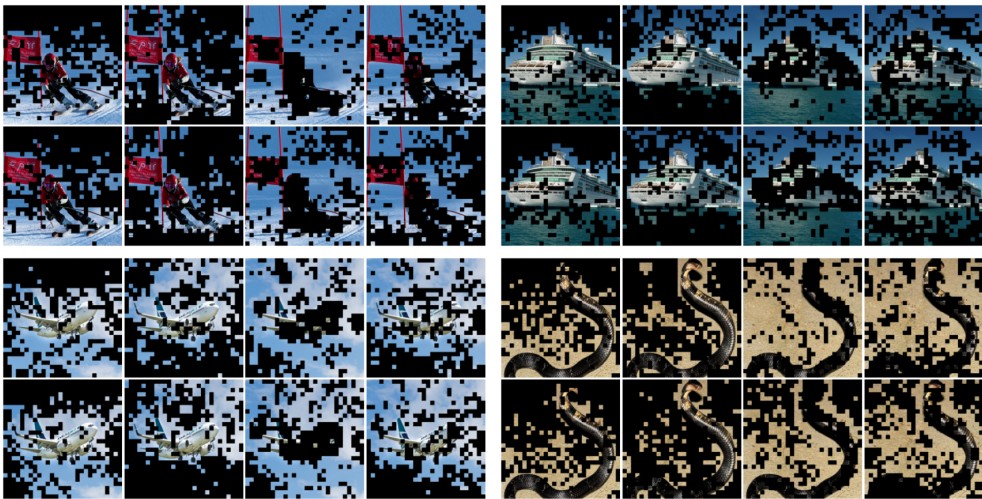

Figure 5: Visualization of routers as masks. The mask position (black) indicates router of 0 here and the background image is the same for each subplot. We show the dynamic sparsity of spiking router for different experts (horizontal direction) and time steps (vertical direction).

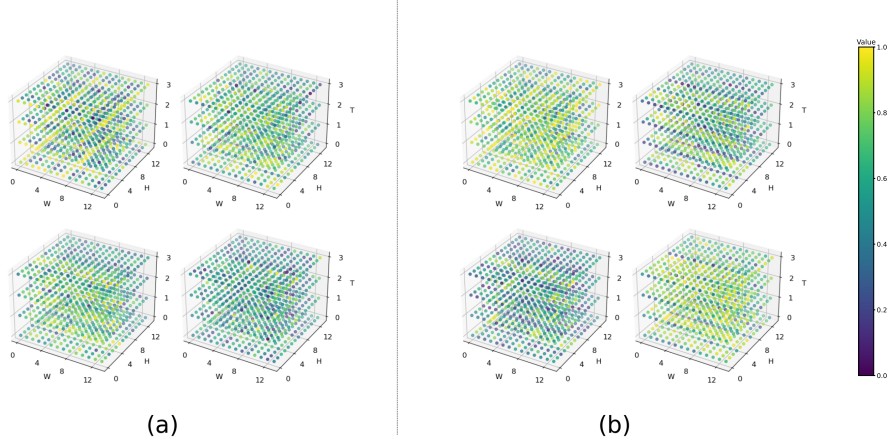

(a)             (b)

Figure 6: Average spiking rate of different kinds of images in the ImageNet validation set in the spatial-temporal dimension. The height of the cube is the time step. (a) Japanese spaniel. (b) Volcano.

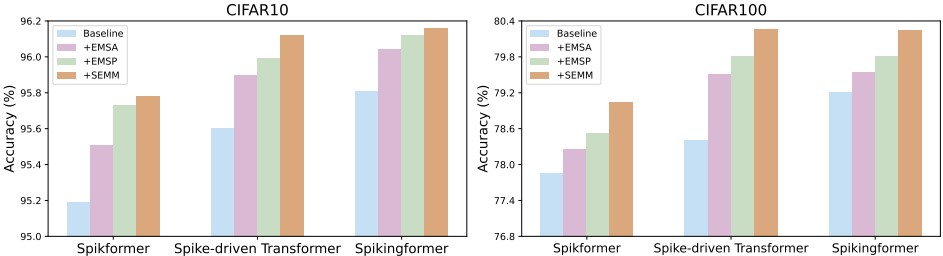

Figure 7: Ablation study of EMSA and EMSP module.

The experimental results on CIFAR, CIFAR10-DVS, and DVS128 Gesture are shown in Tab. 4. These four datasets are relatively small. We basically keep the experimental setup in [11, 12, 14], including the network structure, training settings, etc., and details are given in the Appendix. C.1. SEMM has demonstrated stable performance improvements across various datasets when integrated

Table 3: Results on ImageNet-1k. Model-L-D represents a model with $L$ encoder blocks and $D$ channels.

| Methods | Architecture | Param (M) | Time Step | Top-1 Acc (%) |
|---|---|---|---|---|
| SEW ResNet[7] | SEW-ResNet-34 | 21.79 | 4 | 67.04 |
| | SEW-ResNet-101 | 44.55 | 4 | 68.76 |
| | SEW-ResNet-152 | 60.19 | 4 | 69.26 |
| MS-ResNet[9] | MS-ResNet-18 | 11.69 | 6 | 63.10 |
| | MS-ResNet-34 | 21.80 | 6 | 69.42 |
| | MS-ResNet-104* | 77.28 | 5 | 76.02 |
| Spikformer[11] | Spikformer-8-384 | 16.81 | 4 | 70.24 |
| | Spikformer-8-512 | 29.68 | 4 | 73.38 |
| Spikformer + SEMM | Spikformer-8-384 | **16.05** | 4 | **72.86** |
| | Spikformer-8-512 | **28.22** | 4 | **75.93** |
| Spike-driven Transformer[12] | SD-Transformer-8-384 | 16.81 | 4 | 72.28 |
| | SD-Transformer-8-512 | 29.68 | 4 | 74.57 |
| SD-Transformer + SEMM | SD-Transformer-8-384 | **16.05** | 4 | **73.93** |
| | SD-Transformer-8-512 | **28.22** | 4 | **76.62** |
| Spikingformer[14] | Spikingformer-8-384 | 16.81 | 4 | 72.45 |
| | Spikingformer-8-512 | 29.68 | 4 | 74.79 |
| Spikingformer + SEMM | Spikingformer-8-384 | **16.05** | 4 | **73.58** |
| | Spikingformer-8-512 | **28.22** | 4 | **76.03** |

Table 4: Results on CIFAR10-DVS, DVS128 Gesture, and CIFAR.

| Methods | CIFAR10-DVS | | DVS128 Gesture | | CIFAR-10 | | CIFAR-100 | |
|---|---|---|---|---|---|---|---|---|
| | $T$ | Acc | $T$ | Acc | $T$ | Acc | $T$ | Acc |
| tdBN [8] | 10 | 67.80 | 40 | 96.90 | 6 | 93.20 | - | - |
| PLIF [44] | 20 | 74.80 | 20 | 97.60 | 8 | 93.50 | - | - |
| DIET-SNN [45] | - | - | - | - | 5 | 92.70 | 5 | 69.70 |
| Dspike [46] | 10 | 75.40 | - | - | 6 | 94.30 | 6 | 74.20 |
| DSR [47] | 10 | 77.30 | - | - | 20 | 95.40 | 20 | 78.50 |
| Spikformer [11] | 10 | 78.90 | 10 | 96.90 | 4 | 95.19 | 4 | 77.86 |
| | 16 | 80.90 | 16 | 98.30 | | | | |
| Spikformer + SEMM | 10 | **82.32** | 10 | **97.56** | 4 | **95.78** | 4 | **79.04** |
| | 16 | **82.90** | 16 | **98.63** | | | | |
| Spike-Driven Transformer [12] | 10 | 78.90 | 10 | 96.90 | 4 | 95.60 | 4 | 78.40 |
| | 16 | 80.00 | 16 | 99.30 | | | | |
| Spike-Driven Transformer + SEMM | 10 | **81.10** | 10 | **97.56** | 4 | **96.12** | 4 | **80.26** |
| | 16 | **82.42** | 16 | **99.30** | | | | |
| Spikingformer [14] | 10 | 79.90 | 10 | 96.20 | 4 | 95.81 | 4 | 79.21 |
| | 16 | 81.30 | 16 | 98.30 | | | | |
| Spikingformer + SEMM | 10 | **80.70** | 10 | **96.88** | 4 | **96.16** | 4 | **80.24** |
| | 16 | **82.10** | 16 | **98.56** | | | | |

into different Spiking Transformer baselines. Specifically, SEMM achieves SOTA on CIFAR-10 (96.16%), CIFAR-100 (80.26%), CIFAR10-DVS (82.9%) and DVS128 Gesture (99.3%).

## 4.3 Ablation Study and Hyperparameter Sensitivity

**Module ablation.** EMSA and EMSP together improve the performance of the baseline, as shown in Fig. 7. The use of both EMSA and EMSP alone is better than baseline, illustrating their respective superiority.

**Experts number.** We examine the utility brought by different numbers of experts of EMSA. The results of Spiking Transformer baselines on CIFAR are presented in Fig. 8. It indicates that within a certain range of expert numbers, the results can still be competitive and robust. Among these, we select 4 experts as the parameter setting for our final application.

# 5  Conclusion

In this work, we explored the feasibility of adapting MoE in Spiking Neural Networks and formulated an SNN-MoE paradigm named the Spiking Experts Mixture Mechanism. Unlike the vanilla MoE, which uses softmax and TopK hard sparse, SEMM implements dynamic conditional computation from the viewpoint of spiking sparse activation. With redesigned Router-Expert pairs and element-wise spike-driven operations, SEMM is computation-efficient and SNN-compatible. Embedded in Spiking Transformers, SEMM-based EMSA and EMSP can bring stable performance improvement on static and neuromorphic datasets. SEMM paradigm can inspire future exploration of high-performance, high-capacity Spiking Transformers. We hope SEMM can bring vitality to dynamic conditional computation and the design of next-generation architecture for SNNs. Future work will explore SEMM implementation in a wider range of tasks and larger SNN models.

# 6  Acknowledgments and Disclosure of Funding

We sincerely thank Dr. Wei Fang for his assistance with this work. This work is supported by grants from the National Natural Science Foundation of China (62088102, 62202014, 62332002, 62425101, 62236009, U22A20103, 62441606), Shenzhen Basic Research Program (No.JCYJ20220813151736001), the major key project of the Pengcheng Laboratory (PCL2021A13) and National Science Foundation for Distinguished Young Scholars (62325603). Computing support was provided by Pengcheng Cloud Brain.

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

# A  Overall architecture of Spike-driven Transformer/Spikingformer with SEMM

We provide a detailed description of the overall structure of Spike-driven Transformer with SEMM, and Spikingformer with SEMM. Given a 2D image sequence $\mathbf{I} \in \mathbb{R}^{T \times C \times H \times W}$, the Patch Splitting Module (PSM), consisted of three Convolution-Batch Normalization-$\mathcal{SN}$-Maxpooling layers and one Convolution-Batch Normalization-Maxpooling layer, is used to projects and splits $\mathbf{I}$ into a sequence of $N$ flattened spike patches $\mathbf{s}$ with $D$ channels. The Relative Position Embedding (RPE) module, a Convolution-BatchNorm layer, is used to generate position embedding to get $\mathbf{U}_0$. $\mathcal{SN}(\cdot)$ means the spike neuron layer. Then the $\mathbf{U}_0$ is passed to the $L$-blocks encoder. The encoder block consists of an Experts Mixture Spiking Attention (EMSA) and an Experts Mixture Spiking Perceptron (EMSP) block. Membrane-shortcut residual connections are applied in both the EMSA and EMSP blocks. A global average-pooling (GAP) is utilized on the processed feature from the encoder and outputs the $D$-dimension feature which will be sent to the fully-connected-layer classification head (CH) to output the prediction $Y$. The architecture can be written as follows:

$$\mathbf{u} = \mathrm{PSM}\,(\mathbf{I})\,, \qquad\qquad \mathbf{I} \in \mathbb{R}^{T \times C \times H \times W}, x \in \mathbb{R}^{T \times N \times D}, \tag{28}$$

$$\mathbf{s} = \mathcal{SN}(\mathbf{u}), \qquad\qquad \mathbf{s} \in \mathbb{R}^{T \times N \times D} \tag{29}$$

$$\mathbf{RPE} = \mathrm{BN}(\mathrm{Conv2d}(\mathbf{s})), \qquad\qquad \mathrm{RPE} \in \mathbb{R}^{T \times N \times D} \tag{30}$$

$$\mathbf{U}_0 = \mathbf{u} + \mathbf{RPE}, \qquad\qquad \mathbf{U}_0 \in \mathbb{R}^{T \times N \times D} \tag{31}$$

$$\mathbf{S}_0 = \mathcal{SN}(\mathbf{U}_0), \qquad\qquad \mathbf{S}_0 \in \mathbb{R}^{T \times N \times D} \tag{32}$$

$$\mathbf{U}_l^{'} = \mathrm{EMSA}(\mathbf{S}_{l-1}) + \mathbf{U}_{l-1}, \qquad\qquad \mathbf{U}_l^{'} \in \mathbb{R}^{T \times N \times D}, l = 1...L \tag{33}$$

$$\mathbf{S}_l^{'} = \mathcal{SN}(\mathbf{U}_l^{'}), \qquad\qquad \mathbf{S}_l^{'} \in \mathbb{R}^{T \times N \times D}, l = 1...L \tag{34}$$

$$\mathbf{S}_l = \mathcal{SN}(\mathrm{EMSP}(\mathbf{S}_l^{'}) + \mathbf{U}_l^{'}), \qquad\qquad \mathbf{S}_l \in \mathbb{R}^{T \times N \times D}, l = 1...L \tag{35}$$

$$\mathbf{Y} = \mathrm{CH}(\mathrm{GAP}(\mathbf{S}_L)), \tag{36}$$

Following the Spiking Experts Mixture Mechanism and Spike-driven Self-Attention (SDSA) used in Spike-driven Transformer, the EMSA can be written as:

$$\mathbf{Q} = \mathcal{SN}_{\mathbf{Q}}(\text{L-BN}_{\mathbf{Q}}(\mathbf{X})), \mathbf{K} = \mathcal{SN}_{\mathbf{K}}(\text{L-BN}_{\mathbf{K}}(\mathbf{X})), \mathbf{V} = \mathcal{SN}_{\mathbf{V}}(\text{L-BN}_{\mathbf{V}}(\mathbf{X})), \tag{37}$$

$$\mathbf{A} = \mathrm{SDSA}(\mathbf{Q}, \mathbf{K}, \mathbf{V}) = \mathcal{SN}\,(\mathrm{SUM_c}\,(\mathbf{Q} \odot \mathbf{K})) \odot \mathbf{V}, \tag{38}$$

The rest operate according to standard EMSA 18. The EMSP in Spike Driven Transformer is also much the same as the standard 22, except that no spiking neuron layer in the output layer.

# B  Computation Overhead Details of SEMM

## B.1  EMSA

The linear layers that generates $m$ Query $\mathbf{Q_m}$ have $m(\frac{D^2}{m}) = D^2$ parameters. The linear layer for Key $\mathbf{K}$ has $\frac{D^2}{m}$ parameters. Generating Value $\mathbf{V}$ requires $dD$ parameters. The output linear has $dD$ parameters. The router linear layer transforms $D$ to expert number $m$, and has $mD$ parameters. The number of total parameters is $(1 + 1/m)D^2 + (2d + m)D$. For operations, obtaining Query and Key needs $(1 + 1/m)\overline{R}TND^2$. Computing Value or the output of EMSA requires $d\tilde{R}TND$. The number of operations between $\mathbf{Q_m}, \mathbf{K}, \mathbf{V}$ is $(D + md)R_{\mathcal{R}}TN^2$. The number of router matrix computation operations is $m\tilde{R}TND$. Due to the sparsity of SEMM and the smaller data dimensions, the number of element-wise operations for the router and expert is negligible.

## B.2  EMSP

The parameter number of $\mathbf{W_1}, \mathbf{W_{\mathcal{R}}}, \mathbf{W_o}$ are all $(8//3)D^2$. The number of parameters for $3 \times 3$ depthwise convolution is $3 \times 3 \times (8//3)D = 24D$. The all parameter number is $8D^2 + 24D$. The operation number of $\mathbf{W_1}, \mathbf{W_{\mathcal{R}}}, \mathbf{W_o}$ are all $(8//3)TND^2$. The number of operations for depthwise convolution is $24\tilde{R}TND$.

## C  Experiment Details

### C.1  Experimental Setup

Our experimental framework closely aligns with the methodology outlined in [11]. All of Baseline's code is publicly available, and we abide by their credentials. For the ImageNet-1K, we utilize a fixed timestep count of $T = 4$. The optimizer is the AdamW, with a batch size of 128 or 256 over the course of 310 training epochs. The learning rate is governed by a cosine-decay schedule, starting from an initial value of 0.0004. We incorporate a suite of standard data augmentation techniques, including random augmentation, mixup, and cutmix, into our training. For four small datasets, we adapt the SEMM to a variety of baseline models, following the precedents set by [11, 12]. For CIFAR, we maintain a timestep count of $T = 4$. For the neuromorphic datasets, we increase this to $T = 10$ and $T = 16$, respectively. Our experimental setup is consistent with each Spiking Transformer baselines, as detailed below.

**Spikformer with SEMM.** The training epoch is set to 310 for CIFAR, 200 for DVS128 Gesture, and 106 for CIFAR10-DVS. The batch size is 128 for CIFAR, 16 for DVS128 Gesture and CIFAR10-DVS. The learning rate is initialized to 0.0005 for CIFAR10/100, 0.001 for DVS128 Gesture and CIFAR10-DVS. All of them are reduced with cosine decay. We follow [11] to apply data augmentation on DVS128 Gesture and CIFAR10-DVS. In addition, the network structures used in CIFAR-10, CIFAR10-DVS, and DVS128 Gesture are Spiking Transformer-4-384, Spiking Transformer-2-256 and Spiking Transformer-2-256.

**Spike Driven Transformer with SEMM.** The training epoch is set to 210 for CIFAR10/100 datasets, 200 for DVS128 Gesture, and 106 for CIFAR10-DVS. The batch size is 32 for CIFAR10/100, 16 for DVS128 Gesture and CIFAR10-DVS. The learning rate is initialized to 0.0005 for CIFAR10/100, 0.0003 for DVS128 Gesture and 0.01 for CIFAR10-DVS. The rest is consistent with Spikformer.

**Spikingformer with SEMM** The training epoch is set to 410 for CIFAR10/100 datasets, 200 for DVS128 Gesture, and 106 for CIFAR10-DVS. The batch size is 64 for CIFAR10/100, 16 for DVS128 Gesture and CIFAR10-DVS. The learning rate is initialized to 0.0005 for CIFAR10/100, 0.1 for DVS128 Gesture and CIFAR10-DVS. The rest is consistent with Spikformer.

### C.2  Model Details

In our experiments, we use 8 NVIDIA-4090 GPUs for ImageNet, and 1 NVIDIA-4090 GPU for other datasets. We adjust the value of membrane time constant $\tau$ in spike neuron when training models on DVS datasets. In direct training SNN models with surrogate function,

$$Sigmoid(x) = \frac{1}{1 + \exp(-\alpha x)} \tag{39}$$

We select the Sigmoid function as the surrogate function with $\alpha = 4$ in all experiments.

### C.3  Additional Experiments

Table 5: Ablation study results on the time step.

| Datasets | Models | Time-Step | Top1-Acc(%) |
|---|---|---|---|
| CIFAR10/100 | Spikformer + SEMM | 1 | 94.11/74.67 |
| | | 2 | 94.87/78.49 |
| | | 4 | 95.78/79.04 |
| | | 6 | 95.99/79.29 |
| | Spike Driven Transformer + SEMM | 1 | 94.97/76.77 |
| | | 2 | 95.58/78.49 |
| | | 4 | 96.12/80.26 |
| | | 6 | 96.48/80.87 |
| | Spikingformer + SEMM | 1 | 94.54/77.16 |
| | | 2 | 95.42/78.85 |
| | | 4 | 96.16/80.24 |
| | | 6 | 96.67/81.22 |

The accuracy regarding different simulation time steps of the spiking neuron is shown in Tab. 5. At all the time steps, our Spiking Transformer with SEMM shows competitive performance. When the

time step is 1, our Spikformer with SEMM is 1.7% lower than the network with $T = 4$ on CIFAR10, and our Spikformer with SEMM with 1 time step still achieves 74.67% on CIFAR100. The above results show that Spikformer with SEMM is robust under fewer time steps.

In EMSP, the role of Deep Wise Convolution layer (DWC) is to independently extract features at the channel-level experts using a 3x3 receptive field, thereby enhancing representational capacity. The ablation study about DWC in Tab. 6 demonstrates its effectiveness.

Table 6: Ablation on Deep Wise Convolution layer.

| Model | Module | CIFAR100 | CIFAR10-DVS |
|---|---|---|---|
| Spikformer | EMSP | 78.53 | 82.32 |
| Spikformer | EMSP w/o DWC | 78.17 | 81.30 |
| Spike-driven Transformer | EMSP | 79.81 | 81.10 |
| Spike-driven Transformer | EMSP w/o DWC | 78.95 | 80.51 |
| Spikingformer | EMSP | 79.81 | 81.95 |
| Spikingformer | EMSP w/o DWC | 79.44 | 81.20 |

## D  Visualization

### D.1  Image Samples Visualization

More dynamic allocation visualizations of the spiking router are given in Fig. 9

## E  Discussion on the Feasibility of Hardware Deployment for SEMM

SEMM can theoretically be deployed on chips that are synchronous within layers and asynchronous between layers, such as TrueNorth [48]. Specifically, multiple spiking router matrices, i.e. $\{r_1, r_2, ..., r_m\}$, can be implemented by one block (Linear-BatchNorm-SpikingNeuron), thus can be mapped to the one core of the neuromorphic chip for computation. Outputs from the same core are considered synchronous. The spiking Hadamard product and addition that occur in SEMM are analogous to the AND and ADD operations in SEW ResNet [7]. These element-wise spiking operations can be adapted on neuromorphics chips that support multiple branches and residuals, such as Speck V2 [49].

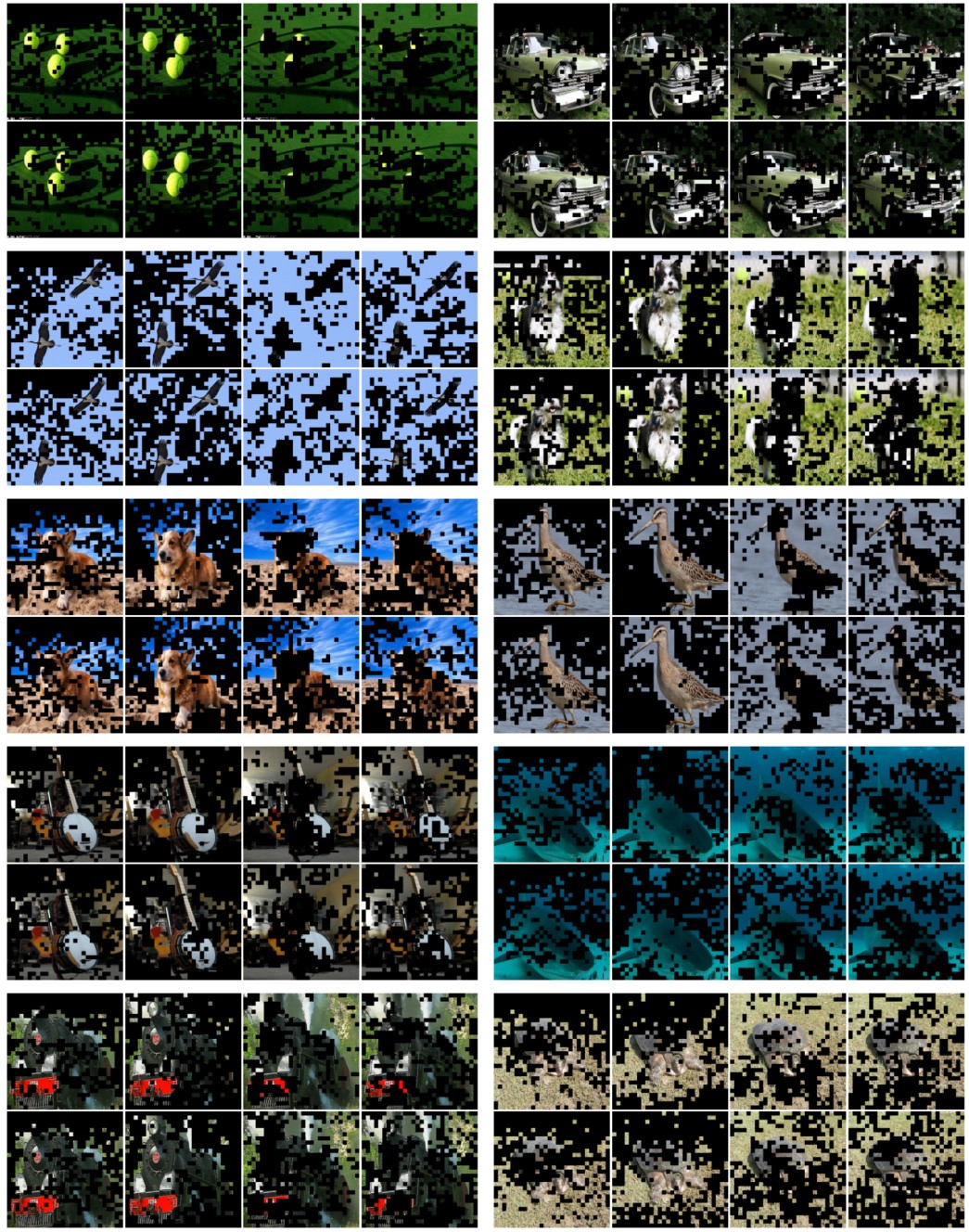

Figure 9: More visualization samples of routers as masks.

