# OpenReview forum: "Spiking Transformer with Experts Mixture"
_NeurIPS.cc/2024/Conference — NeurIPS 2024 poster_

### Official Review · Reviewer_eZ3Z · 2024-07-02

**Soundness:** 3
**Presentation:** 3
**Contribution:** 3
**Rating:** 7
**Confidence:** 5

**Summary:**

The manuscript presents an innovative approach to integrating SNNs and MoE methodologies into a cohesive framework.  It introduces the Spiking Experts Mixture Mechanism (SEMM), which leverages the sparse spiking activations of SNNs and the conditional computation of MoE.  The proposed SEMM has been adapted into the Spiking Transformer architecture, resulting in two novel structures: the Experts Mixture Spiking Attention (EMSA) and the Experts Mixture Spiking Perceptron (EMSP). Experimental results are provided, showing SEMM's effectiveness in achieving stable improvements on neuromorphic and static datasets with manageable increases in computational overhead.

**Strengths:**

1. Novel mechanisms. The introduction of SEMM, along with its derivatives EMSA and EMSP, is a novel contribution, particularly in how these mechanisms manage to perform conditional computation dynamically.
2. The improvement of performance brought by SEMM is significant. The EMSA and EMSP modules can be adapted to any spiking self-attention and MLP modules, which may be beneficial to broader SNN research community.
3. Experiments are comprehensive. Especially, Fig. 6 shows the average spiking rate of spatial-temporal locations of routers in different kinds of images, which proves the effectiveness of spiking routing functions.

**Weaknesses:**

1. Several Unclear Notations.
(1) In Fig.1 (b) and Line 132, the authors use the notation of $N$, but what does the $N$ denote? I cannot find explanations on this issue. Does $N$ denote the length of image patches?
(2) In Fig.2 (c), why does "DWL" stand for Depth-Wise Conv? I think it would be better to denote it as "DWC" or just explain it as Depth-Wise Linear.
(3) In Equation 24 and 26, what’s the meaning of $p$ and $q$?
Please clarify these notations.
2. The Section 4.3 is more like a sensitivity experiment of hyper-parameters rather than an ablation study. The real ablation study has been shown in Fig. 4. Please correct this issue to avoid misleading readers.

**Questions:**

1. Can you discuss the feasibility of adapting your proposed EMSA and EMSP modules to neuromorphic hardware? Are there any specific types of neuromorphic hardware or setups that are particularly well-suited or ill-suited for deploying SEMM?
2. How does SEMM handle the trade-off between sparsity and accuracy?

**Limitations:**

While the authors indicate that the limitations are addressed in Section 5 (refer to Line 533 in the Checklist), it appears that this section does not currently include a discussion on the limitations.
Can you include a discussion on the limitations of the work?

---

> ### Author Rebuttal · Authors · 2024-08-07
>
> Thank you for your comments and suggestions for improvement.
>
> > ***Weakness 1**: Several Unclear Notations.*
>
> **AW1**: Sorry for the confusion caused by unclear notations. To clarify: __(1)__ $N$ denotes the length of image patches. __(2)__ Thanks for your advice. We will change the abbreviation representing the depth-wise convolutional layer from "DWL" to "DWC". __(3)__  The input current matrix (feature) $I[t]$ has a dimension of $P\times Q$, where p and q represent the $p$-th row and $q$-th column, respectively. We will modify them in the manuscript.
>
> > ***Weakness 2**: The Section 4.3 is more like a sensitivity experiment of hyper-parameters rather than an ablation study.*
>
> **AW2**: We appreciate your suggestion and will revise the titles of the ablation study and the hyperparameter sensitivity experiment in the manuscript accordingly.
>
> > ***Question 1**: Can you discuss the feasibility of adapting your proposed EMSA and EMSP modules to neuromorphic hardware?*
>
> **AQ1**:
> SEMM can theoretically be deployed on chips that are synchronous within layers and asynchronous between layers, such as TrueNorth [R1]. The spike-driven operation of SEMM is similar to the AND and ADD residuals in SEW ResNet [R2]. Multiple branches and residuals are supported on certain neuromorphic hardware, such as Speck V2 [R3]. However, deploying SEMM on a fully asynchronous neuromorphic chip is challenging.
>
>
> > ***Question 2**: How does SEMM handle the trade-off between sparsity and accuracy?*
>
> **AQ2**:
> In SEMM, we do not control sparsity using hyperparameters; instead, the model dynamically adjusts based on the input. Therefore, there is no trade-off between sparsity and performance. Higher sparsity does not necessarily imply lower accuracy. SEMM utilizes dynamic spiking sparsity to select feature representations within each expert, thereby enhancing accuracy.
>
> > ***Limitation 1**: While the authors indicate that the limitations are addressed in Section 5 (refer to Line 533 in the Checklist), it appears that this section does not currently include a discussion on the limitations. Can you include a discussion on the limitations of the work?*
>
> **AL1**:
> We apologize for the confusion caused by our incomplete statement. In Sec. 5, we state that future work will focus on SEMM application on larger SNN models. In line with your suggestion, we further discuss the limitations of this work:
>
> SEMM has been validated on image datasets, demonstrating its effectiveness. However, no further verification has been conducted on other tasks and diverse datasets. Despite validating SEMM's effectiveness
> in time series forecasting tasks as suggested by reviewer RYQd, demonstrating its generalizability, there remain numerous tasks yet to be validated. Additionally, the effectiveness of SEMM on larger SNN models has yet to be validated. Future work will address these two limitations.
>
> We will add this discussion to the appendix of the manuscript.
>
> [R1] Akopyan, Filipp, et al. "Truenorth: Design and tool flow of a 65 mw 1 million neuron programmable neurosynaptic chip." IEEE transactions on computer-aided design of integrated circuits and systems 34.10 (2015): 1537-1557.
>
> [R2] Fang, Wei, et al. "Deep residual learning in spiking neural networks." Advances in Neural Information Processing Systems 34 (2021): 21056-21069.
>
> [R3] Ole, Richter, et al. "Speck: A Smart event-based Vision Sensor with a low latency 327K Neuron Convolutional Neuronal Network Processing Pipeline." IEEE International Symposium On Asynchronous Circuits and Systems (2023).

---

> > ### Comment · Reviewer_eZ3Z · 2024-08-09
> > **Good Rebuttal**
> >
> > I have thoroughly reviewed the authors' response, as well as their replies to the other reviewers. The authors have addressed my concerns, and I was pleasantly surprised to find that in their response to Reviewer RYQd, they mentioned applying SEMM to Spikformer, which notably improves time-series forecasting performance. This finding left a strong impression on me. In light of this, I have decided to raise my score.

---

> ### Author Response · Authors · 2024-08-13
>
> Dear Reviewer,
>
> Thank you again for your valuable time and feedback.
>
> Best regards
>
> Authors

---

### Official Review · Reviewer_rWD5 · 2024-07-07

**Soundness:** 2
**Presentation:** 3
**Contribution:** 3
**Rating:** 7
**Confidence:** 4

**Summary:**

The authors introduce MOE into SNN, propose the SEMM structure, and use this structure to enhance the attention and MLP modules in the Spikeformer-like architecture. They transform these elements into the EMSA and EMSP modules. The new network architecture achieved better performance under similar parameter settings compared to the original spiking transformer structures.

**Strengths:**

The SEMM proposed by the authors does not introduce floating-point matrix multiplication during the SNN forward process. Its SMSA structure effectively focuses on the relevant parts of the image. As a result, SEMM enhances the performance of various spiking transformers, achieving SOTA performance on different datasets.

**Weaknesses:**

1. The authors claim that SEMM's advantage lies in its spike-driven characteristics and suitability for asynchronous chips. However, the author did not run the SEMM structure on real asynchronous hardware. In my opinion, due to the spike-driven nature of an asynchronous chip lacking a hardware clock, it is difficult for multiple spike matrices in the router part (in SEMM) to arrive simultaneously and interact, resulting in most outputs being 0. The author needs to provide a more detailed explanation of the spike-driven advantage, especially on asynchronous hardware.

2. Prior to Eqn. 14, the authors omitted the equation for partitioning the input X (how to get the $A_m$?).

3. In Eqn. 17, $BN(SEMM(E, R, F(·)))W_o$ undoubtedly involves floating-point matrix operations; is there a writing error present here?

**Questions:**

1. The SEMM(E, R, F(·)) obtained from Eqn. 16 is described as containing only 0 or 1. But it should be a floating-point matrix.

2. Line 217 What means “The ASR of EMSA is around 0.5, which is comparable to the regular TopK setting of ANN-MoE”? What's the relationship between the ASR in EMSA and the TopK rate in ANN-MoE?

3. What is the purpose of Fig. 6? In my view, different inputs leading to different activations are natural.

---

> ### Author Rebuttal · Authors · 2024-08-07
>
> We appreciate your detailed comments. We would like to address your concerns below.
>
> > ***Weakness 1**: The author needs to provide a more detailed explanation of the spike-driven advantage, especially on asynchronous hardware.*
>
> **AW1**: As you mentioned, deploying SEMM on a fully asynchronous neuromorphic chip is challenging. However, SEMM can theoretically be deployed on chips that are synchronous within layers and asynchronous between layers, such as TrueNorth [R1]. Specifically, multiple spiking router matrices, i.e. $\{ r_1,r_2,...,r_m\}$, can be implemented by one block (Linear-BatchNorm-SpikingNeuron) and thus can be mapped to the one core of the neuromorphic chip for computation. Outputs from the same core are considered synchronous. The spiking Hadamard product and addition that occur in SEMM are analogous to the AND and ADD operations in SEW ResNet [R2]. These element-wise spiking operations can be adapted on the neuromorphic chip that supports multiple branches and residuals, such as Speck V2 [R3].
>
>
> > ***Weakness 2**: Prior to Eqn. 14, the authors omitted the equation for partitioning the input X (how to get the $A_m$?).*
>
> **AW2**: Sorry for the confusion caused by ommitation. In Eqn. 14, Each $\bf A_m$ is obtained as follows,
> $$
> {\bf A\_m} = \text{SSA}(\bf Q\_m,K,V),
> $$
> where $\bf Q_m$ is individual to each expert, and $\bf K, V$ are shared among experts. We will revise it in the manuscript.
>
>
> > ***Weakness 3, Question 1**: In Eqn. 17, $BN(SEMM(E, R, F(·)))W_o$ undoubtedly involves floating-point matrix operations; is there a writing error present here? The SEMM(E, R, F(·)) obtained from Eqn. 16 is described as containing only 0 or 1. But it should be a floating-point matrix.*
>
> **AW3**: We apologize for our incorrect expression. The $\text{F}(·)$ in Eqn. 16 involves spiking Hadamard products and element-wise addition. Since both $\bf E$ and $\mathcal{R}$ are in spiking form, the output of ${\rm{SEMM}}({\bf E}, \mathcal{R}, {\text F}(\cdot))$ is an integer spiking form rather than a binary spiking form. $\text{BN}({\rm{SEMM}}({\bf E}, \mathcal{R}, {\text F}(\cdot)))\bf W_o$  in Eqn. 17 can be decomposed into addition operations and multiplication is avoided, which is shown as
> $$ {\rm{SEMM}}({\bf E}, \mathcal{R}, {\text F}(\cdot)) = \sum_{i=1}^{m} {\bf r}_i * {{\bf A}_i} = \bf s_1 + s_2 + \cdots + s_m $$
> $$ {\rm{SEMM}}({\bf E}, \mathcal{R}, {\text F}(\cdot))\bf W_o = ({\bf s_1 + s_2 + \cdots + s_m})\bf W_o = \bf s_1W_o + \bf s_2W_o + \cdots + \bf{s_m}W_o$$
> where $\bf s_m$ is binary spiking form. We omit $\text{BN}$ because its parameters can be integrated with the linear layer. We will revise it in the manuscript.
>
> >**Question 2**: Line 217 What means "The ASR of EMSA is around 0.5, which is comparable to the regular TopK setting of ANN-MoE"? What's the relationship between the ASR in EMSA and the TopK rate in ANN-MoE?
>
> **AQ2**:  In ANN-MoE [R4, R5], TopK typically selects 2 experts, which can be considered as 50% sparsity for a total of four experts in our settings. This is similar to the average spiking rate (sparsity) observed in EMSA.
>
> >**Question 3**: What is the purpose of Fig. 6? In my view, different inputs leading to different activations are natural.
>
> **AQ3**:  The purpose of Fig. 6 is to further demonstrate the dynamic condition computing characteristics of routers in SEMM. It illustrates the average spiking rate across spatial-temporal locations of routers across different classes of images (with 50 images per class), offering a more representative visualization compared to the single image in Fig. 5. Specifically, the spiking pattern of a router varies with time steps and spatial positions. Across different categories, the active routers differ; for example, in (a), routers 1 and 2 (top left and top right) are active, while in (b), routers 1 and 4 (top left and bottom right) exhibit high activation.
>
> [R1] Akopyan, Filipp, et al. "Truenorth: Design and tool flow of a 65 mw 1 million neuron programmable neurosynaptic chip." IEEE transactions on computer-aided design of integrated circuits and systems 34.10 (2015): 1537-1557.
>
> [R2] Fang, Wei, et al. "Deep residual learning in spiking neural networks." Advances in Neural Information Processing Systems 34 (2021): 21056-21069.
>
> [R3] Ole, Richter, et al. "Speck: A Smart event-based Vision Sensor with a low latency 327K Neuron Convolutional Neuronal Network Processing Pipeline." IEEE International Symposium On Asynchronous Circuits and Systems (2023).
>
> [R4] Noam Shazeer, et al. "Outrageously large neural networks: The sparsely-gated mixtureof-experts layer." International Conference on Learning Representations (2017).
>
> [R5] Yanqi Zhou, et al. "Mixture-of-experts with expert choice routing." Advances in Neural Information Processing Systems (2022).

---

> > ### Comment · Reviewer_rWD5 · 2024-08-09
> >
> > Thanks for your response. All my concerns have been addressed.

---

> ### Author Response · Authors · 2024-08-13
>
> Dear Reviewer,
>
> Thank you for your recognition of this work.
>
> Best regards
>
> Authors

---

### Official Review · Reviewer_RYQd · 2024-07-11

**Soundness:** 3
**Presentation:** 2
**Contribution:** 2
**Rating:** 3
**Confidence:** 5

**Summary:**

The paper presents a novel integration of Spiking Neural Networks (SNNs) with Mixture-of-Experts (MoE) to form the Spiking Transformer, introducing the Spiking Experts Mixture Mechanism (SEMM). The SEMM enables dynamic sparse-conditional computation by having both experts and routers output spiking sequences, which aligns with the sparse and energy-efficient nature of SNNs. The proposed model incorporates two key components: Experts Mixture Spiking Attention (EMSA) for head-wise routing and Experts Mixture Spiking Perceptron (EMSP) for channel-wise allocation of spiking experts. Empirical evaluations demonstrate that SEMM improves performance on neuromorphic and static datasets with minimal additional computational overhead compared to traditional SNN-based transformers.

**Strengths:**

1.Originality: The fusion of SNNs with MoE principles is innovative and provides a fresh perspective on conditional computation.

2.Quality: The paper offers a rigorous mathematical and conceptual framework for SEMM, demonstrating a thorough understanding of both SNNs and MoE.

3.Clarity: While the concepts are complex, the authors have made efforts to explain them coherently, making the paper accessible to a wide audience.

4.Significance: The work has the potential to impact energy-efficient AI, particularly in edge devices where power consumption is a primary concern.

**Weaknesses:**

1.The paper could benefit from more comprehensive empirical evaluations across diverse datasets to generalize the findings.

2.The lack of theoretical guarantees or analysis regarding the convergence and stability of SEMM leaves room for skepticism about its robustness.

3.A more detailed discussion on computational efficiency and scalability with respect to dataset size and complexity would be beneficial.

**Questions:**

1.Can the authors provide more empirical evidence demonstrating the effectiveness of SEMM on larger and more diverse datasets?
2.How does the proposed SEMM adapt to varying levels of sparsity in the input data, and what is the impact on computational efficiency?

**Limitations:**

1.Generalizability: The experiments, though indicative of the SEMM's effectiveness, are limited to a specific subset of datasets. Extending the evaluation to include a wider variety of data, especially those with different characteristics and scales, would strengthen claims about the model's versatility and robustness.
2. Implementation Details for Reproducibility: Although the paper outlines the conceptual framework of SEMM, more explicit details on the implementation and the source code, such as hyperparameter tuning strategies and specifics of the spiking sequences' generation and handling, would facilitate reproducibility and foster further advancements built upon this work.
3. The novelty of the router mechanism for expert mixture is limited for the router mechanism seems have little difference with that in ANN MoE methods.

---

> ### Author Rebuttal · Authors · 2024-08-07
>
> Thanks for your valuable comments. We will explain and discuss your concerns.
>
> > ***Weakness 1, Question 1, Limitation 1**: More experiments across diverse datasets to generalize the findings.*
>
> **A1**: __These datasets in experiments possess distinct characteristics and scales, which verify the effectiveness and generalization capability of SEMM.__ Specifically, the neuromorphic dataset exhibits temporal characteristics and sparse event features, validating SEMM's capability to handle sparse event data. The ImageNet-1K, with its 1,000 classes, verifies SEMM's ability to process complex large-scale data.
> __In terms of data scale,__ we apologize for not being able to provide results on larger datasets due to the rebuttal time and resource constraints.
> __In terms of data diversity,__ we add the validity and generalization experiments of SEMM on time series forecasting, which is a challenging task and aims to predict future values based on historical observations arranged chronologically. Addressing this task often involves modeling the temporal dynamics, resonating profoundly with the nature of neural coding and SNN. As shown in Tab. R2-1, SEMM demonstrates superior predictive capabilities over the baseline on four datasets, further illustrating its versatility and robustness.
> ### Table R2-1
> | Model   |Horizon    | Metr-la [R1]   | Pems-bay [R1] |  Solar [R2]  | Electrictiy [R2]   |
> |---------|:---------:|---------|---------|---------|---------|
> |  |  | $\text R^2$ / RSE | $\text R^2$ / RSE | $\text R^2$ / RSE  | $\text R^2$ / RSE |
> | Spikformer |6 | 0.713/ 0.565 | 0.773/ 0.514 | 0.929/ 0.272| 0.959/ 0.373 |
> | Spikformer+SEMM |  6  | 0.721/ 0.557 | 0.776/ 0.510  | 0.930/ 0.270| 0.964/ 0.338 |
> | Spikformer |24 | 0.527/ 0.725 | 0.697/ 0.594 | 0.828/ 0.426| 0.955/ 0.371 |
> | Spikformer+SEMM |  24 | 0.534/ 0.719 | 0.707/ 0.584 | 0.835/ 0.423  | 0.968/ 0.363 |
> | Spikformer |48| 0.399/ 0.818 | 0.686/ 0.606 | 0.744/ 0.519| 0.955/ 0.379 |
> | Spikformer+SEMM | 48| 0.400/ 0.811 | 0.688/ 0.603 | 0.751/ 0.464 |0.968/ 0.320 |
> | Spikformer |96 | 0.267/ 0.903 | 0.667/ 0.621 | 0.674/ 0.586| 0.954/ 0.382 |
> | Spikformer+SEMM |  96 | 0.279/ 0.895 | 0.673/ 0.618 | 0.675/ 0.581 | 0.961/ 0.381 |
>
> Here, the horizon represents the prediction length. RSE stands for Root Relative Squared Error, where a smaller value indicates a lower error. $\text R^2$ represents the coefficient of determination, with higher values indicating higher prediction accuracy.
>
> > ***Weakness 2**: The lack of theoretical guarantees or analysis regarding the convergence and stability of SEMM leaves room for skepticism about its robustness.*
>
> **AW2**: We regret that we are unable to provide a theoretical analysis of the convergence and robustness of SEMM. To the best of our knowledge, there are currently no works theoretically analyzing the convergence of deep SNNs. __However, to address your concern, we validate SEMM's convergence and robustness through experiments.__
>
> __Convergence: From an experimental perspective, SEMM does not affect the final convergence results of the spiking transformer.__ In Fig. R2-1 and  R2-2 of our submitted PDF rebuttal document, we compare the training loss and test accuracy curves between the Spikformer baseline and SEMM. The introduction of SEMM did not affect convergence.
>
> __Robustness: From an experimental perspective, SEMM does not compromise the robustness of the model.__ To show this, we add experiments about input noise on  CIFAR100. We add a Gaussian noise with mean 0 and variance $\sigma^2$ on inputs $X$. The results are shown in Tab. R2-2. It can be found that at each noise level, introducing SEMM is still better than the baseline.
> ### Table R2-2
> | Model |$\sigma= 0$ (no noise) |$\sigma= 0.1$|$\sigma= 0.2$|$\sigma= 0.3$|
> | -------- |:--------:|:--------:|:--------:|:--------:|
> | Spikfromer  | 77.86    | 72.62   | 70.37 | 68.39|
> | Spikformer with SEMM  | 79.04 | 73.47|71.05|69.54|
>
> > ***Weakness 3**: A more detailed discussion on computational efficiency and scalability with respect to dataset size and complexity would be beneficial.*
>
> **AW3**: Thanks for your comments. Following your suggestion, we show that SEMM exhibits higher computational efficiency compared to ANN-MoE paradigm on three diverse datasets, i.e., CIFAR100 (static images), CIFAR10-DVS (neuromorphic images), and  Solar (time series data). As shown in Tab.R2-3 (following), the number of parameters (Param) and computational operations (OPs) of SEMM on the three datasets are both lower than that of ANN-MoE, while also achieving the best performance. This demonstrates its computational efficiency and scalability for different datasets.
> ### Table R2-3
> | Model | MoE-Type|CIFAR100 | CIFAR10-DVS | Solar |
> | -------- |:--------|:--------:|:--------:|:--------:|
> |   | | Param(M)/OPs (G)/ Acc (%) |Param(M)/OPs (G)/ Acc (%)|Param(M)/OPs (G)/ $\text R^2$|
> | Spikformer  |Baseline | 9.32/0.83/77.86 |2.57/1.06/80.90|2.52/0.88/0.828
> | Spikformer  |SEMM | 8.98/0.94/79.04 |2.57/0.89/82.90|2.52/0.78/0.835
> | Spikformer  |AM-F  | 23.60/1.22/77.35 |5.74/1.45/80.80|5.69/1.16/0.821
> | Spikformer  |AM-S  | 23.60/1.34/76.79 |5.74/1.42/82.90|5.69/1.05/0.819
>
> > ***Question 2**: How does the proposed SEMM adapt to varying levels of sparsity in the input data, and what is the impact on computational efficiency?*
>
> **AQ2**: __Regarding question 2, could you further explain what you mean by data sparsity?__
>
> > ***Limitation 2**: Implementation Details for Reproducibility.*
>
> **AL2**: You can refer to the Appendix. C for the hyperparameters and find the source code of SEMM in the supplementary materials.
>
> [R1] Li, Yaguang, et al. "Diffusion convolutional recurrent neural network: Data-driven traffic forecasting." arXiv preprint arXiv:1707.01926 (2017).
>
> [R2] Lai, Guokun, et al. "Modeling long-and short-term temporal patterns with deep neural networks." The 41st international ACM SIGIR conference on research & development in information retrieval (2018).

---

> > ### Comment · Reviewer_RYQd · 2024-08-13
> > **comments**
> >
> > Thank you for your response and for supplementing the experiments with the time series dataset. However, there are still some concerns that remain unresolved:
> >
> > 1.The question regarding varying sparsity stems from the statement in your paper that "SEMM has a variable sparsity when dealing with different data" (Column 141, Page 4).
> >
> > 2.In Table R2-3, the parameter count and operations (OPs) do not appear to show significant advantages over the Spikformer baseline. In fact, the OPs of Spikformer SEMM are even higher than those of the Spikformer baseline. This raises doubts about the claimed advantage of "realizes sparse conditional computation" (as stated in the abstract). That is, what is the final contributions or  advantages of  " sparse conditional computation"? That still is confused.
> >
> > 3.The efficiency of the expert mixture system based on the Spiking Transformer has not been evaluated on larger datasets, which is a limitation that cannot be ignored. Models with expert mixture architectures typically demonstrate superior performance on larger datasets, so a broader evaluation would be necessary to support the high quality publication for  this conference.
> >
> > 4.There are also concerns regarding the novelty of the proposed approach. Specifically, the differences between the proposed model and existing ANN MoE methods beyond the LIF neurons are not so strong to support the nolvelty of the proposed model, as mentioned by other reviewers.

---

> ### Author Response · Authors · 2024-08-13
>
> Dear Reviewer,
>
> We hope this message finds you well.
> We have provided thorough responses to you and sincerely hope you can look through them and update the scores if your concerns have been resolved. We are also open to further discussion if the concerns have not been fully addressed. Please feel free to let us know if you still have any questions.
>
> Best regards
>
> Authors

---

> ### Author Response · Authors · 2024-08-14
> **Rebuttal by Authors**
>
> Dear Reviewer,
> Thank you for your time and effort in reviewing our work. We will explain and discuss your concerns.
>
> > ***Concern 1**: Explain the statement "SEMM has a variable sparsity when dealing with different data".*
>
> **AW1**: Sorry for the confusion caused by the unclear statement. We intend to convey that SEMM exhibits variable sparsity when dealing with different data within the same dataset. As shown in Fig. 5,6, and Tab. 2, for different images, each expert's corresponding router selects a different number and position of tokens, while the overall low firing rate indicates the sparsity.
>
> > ***Concern 2**: Show the final contributions or advantages of "sparse conditional computation".*
>
> **AW2**: Similar to ANN-MoE mechanisms, the sparsity conditional computation characteristic of SEMM serves as the source of the performance improvement and computational efficiency achievement. Our experiments show that SEMM can bring stable performance improvements to SNN Transformers.
>
> In terms of computational efficiency, as shown in Tab.R2-3, SEMM results in a substantial reduction in both parameters and operations (OPs), approaching the level of the baseline when compared to the introduction of ANN-MoE mechanisms.
>
> > ***Concern 3**: Evaluation on larger datasets.*
>
> **AW3**: We apologize for not being able to provide results on larger datasets due to the resource constraints. We will add discussions on this issue in the Limitation.
>
> > ***Concern 4**: Only the LIF neurons are not so strong to support the novelty of the proposed model.*
>
> **AW4**: Unlike ANN-MoE, the spike-form routers and experts in SEMM, as well as their integration process, are all in line with the spike-driven characteristics of SNNs. At the same time, SEMM also achieves a similar effect of sparse conditional computation as ANN-MoE, thereby stably enhancing performance on SNN Transformers. In addition, the lightweight architectural design is also an advantage over ANN-MoE. As shown in various experiments, the number of parameters and the number of operations after introducing SEMM are close to the baselines, far less than that of ANN-MoE introduction.
>
> Best regards
>
> Authors

---

### Official Review · Reviewer_dEFi · 2024-07-12

**Soundness:** 2
**Presentation:** 3
**Contribution:** 2
**Rating:** 3
**Confidence:** 5

**Summary:**

This research introduces the Spiking Experts Mixture Mechanism (SEMM), a paradigm that combines Spiking Neural Networks (SNNs) with Mixture-of-Experts (MoE) to enhance the capabilities of Spiking Transformers. The proposed SEMM leverages the energy-efficient, sparse spiking activation characteristic of SNNs, resulting in an SNN-MoE framework that is both computation-efficient and SNN-compatible. By integrating SEMM into existing spiking Transformers, performance improvements are achieved. This work contributes to the exploration of high-performance and high-capacity Spiking Transformers.

**Strengths:**

1.	This study is the first attempt to utilize MoE on Spiking Transformers, leveraging the advantages of SNN and MoE for stable performance gains.
2.	The experimental analysis on sparse conditional computation is thorough.
3.	The writing is clear and accessible.

**Weaknesses:**

**Major:**

1.	As mentioned in the first contribution (Line 68-69), the proposed SEMM is a universal SNN-MoE paradigm. Therefore the application of SEMM on the most state-of-the-art Spiking Transformers such as Spike-driven Transformer V2[R1] is crucial to establish the generalizability of SEMM. However, provided baselines do not include it, which weakens the persuasiveness of its generalizability. Moreover, SEMM-based Spiking Transformers underperform compared to [R1]'s peak results, suggesting MoE might not be essential for boosting performance. I would consider raising my rate if you can provide consistent performance improvements using SEMM on [1].
2.	Can you provide evidence or previous studies demonstrating the effectiveness of using DWConv after MLP.fc1? If such evidence exists, it should be cited; otherwise, conducting an ablation study on this layer is recommended. Without this clarification, the motivation behind adding this layer remains unclear, weakening the persuasiveness of SEMM's design applied to MLP.
3.	Is the ANN-MoE paradigm derived from prior ANN-MoE studies, or is it an original concept from your research? If it's based on earlier work, citations are necessary; otherwise, you should highlight the unique advantages of your proposed ANN-MoE paradigm compared to existing ones.

**Minor:**

1.	The description for Figure 3 is not clear enough. Given the explanation provided, I cannot fully understand the meaning of each line in the diagram.
2.	In Figure 4, do the four graphs in each sub-figure relate to four images from the same class?

References:

[R1] Man Yao, JiaKui Hu, Tianxiang Hu, Yifan Xu, Zhaokun Zhou, Yonghong Tian, Bo XU and Guoqi Li.  Spike-driven transformer v2: Meta spiking neural network architecture inspiring the design of next-generation neuromorphic chips. In International Conference on Learning Representations (ICLR), 2024.

**Questions:**

See weakness

**Limitations:**

See weakness

---

> ### Author Rebuttal · Authors · 2024-08-07
>
> Thank you for your detailed comments and suggestions for improvement. We would like to address your concerns and answer your questions below.
>
> > ***Weakness 1**: The application of SEMM on the most state-of-the-art Spiking Transformers such as Spike-driven Transformer V2.*
>
> **AW1**:  Following your suggestion, we report the Spike-driven Transformer V2 (SDT V2) with SEMM on CIFAR100 and ImageNet in Tab. R1-1.
> ### Table R1-1
> | Model | CIFAR100（T=4） | ImageNet (T=1)|
> | -------- |:--------:|:--------:|
> | SDT-V2-15M  | 80.40    | 71.80    |
> | SDT-V2-15M with SEMM  | **81.04**      | **73.14**    |
>
> After configuring SEMM, SDT V2 demonstrates a consistent performance improvement, but the degree of improvement is not as large as that of SDT V1. This could be attributed to SDT V2 being a hybrid spiking CNN-Transformer framework, where the first two stages are based on spiking convolutions, with the third and fourth stages utilizing spiking transformers. Therefore, compared to SDT V1, the proportion of EMSA and EMSP blocks in SDT V2 is lower, resulting in a smaller gain. We will add these results to the manuscript appendix.
>
> Furthermore, we argue that the SEMM differs from the framework design of Spiking Transformers (Spikformer, Spike-driven Transformer V1/2, and Spikingformer, et al.). SEMM is a universal plug-and-play paradigm that can stably enhance performance. It provides a new direction for improving the Spiking Transformer.
>
> > ***Weakness 2**: Provide evidence demonstrating the effectiveness of using DWConv after MLP.fc1.*
>
> **AW2**: In EMSP, the role of DWConv is to independently extract features at the channel-level experts using a 3x3 receptive field, thereby enhancing representational capacity.
> The ablation study about DWConv in Tab. R1-2 demonstrates its effectiveness:
> ### Table R1-2
> | Model | Module |CIFAR100| CIFAR10-DVS|
> | -------- | -------- |:--------:|:--------:|
> | Spikformer  |EMSP  |  78.53   |  82.32  |
> | Spikformer  |EMSP w/o DWConv|  78.17 |  81.30  |
> | Spike-driven Transformer  |EMSP  | 79.81   | 81.10   |
> | Spike-driven Transformer  |EMSP w/o DWConv| 78.95  |  80.51   |
> | Spikingformer  |EMSP  | 79.81  | 81.95   |
> | Spikingformer  | EMSP w/o DWConv| 79.44  | 81.20  |
>
> Thanks for your comments, we will include the ablation experiments in the manuscript appendix.
>
> > ***Weakness 3**: Is the ANN-MoE paradigm derived from prior ANN-MoE studies?*
>
> **AW3**: Thank you for your kind reminder. ANN-MoE represents a paradigm derived from previous works on MoE in ANNs [R1, R2],  integrating multiple experts through routing and hard sparsification. The referenced ANN MoE works in the manuscript adopt this paradigm, for example, __manuscript line 32 "Mixture-of-Experts (MoE) [20; 21] is known for allowing each expert to learn specific tasks or features, showing better performance" and line 89 "The Mixture-of-Experts (MoE) [28; 29] combines the predictions of multiple specialized experts"__. For clarity, we will append citations to these works following the term 'ANN-MoE' in the manuscript.
>
> > ***Weakness 4**: The description for Figure 3 is not clear enough.*
>
> **AW4**: Fig. 3 depicts the parameter number (X-axis) and performance (Y-axis) of various MoE strategies under multiple Spiking Transformer baselines. Each line consists of four points from left to right, representing:
> + __1__. Baseline
> + __2.__ EMSP
> +  __3.__ ANN-MoE paradigm with float-point router matrix (AM-F)
> + __4.__ ANN-MoE paradigm with spiking router matrix (AM-S)
>
> For clarity, we convert Fig. 3 into the following Tab. R1-3. It can be observed that our EMSP achieves better performance with fewer parameters, whereas the application of the ANN-MoE paradigm yields suboptimal results. We will revise Fig. 3 in the manuscript to enhance its readability.
> ### Table R1-3
> | Model | MoE-Type|CIFAR100 Param(M)/ Acc| CIFAR10-DVS Param(M) / Acc|
> | -------- |:--------|:--------:|:--------:|
> | Spikformer  |Baseline | 9.32/77.86 |2.57/80.90
> | Spikformer  |EMSP | 9.39/78.53 |2.58/82.32|
> | Spikformer  |AM-F  | 23.60/77.35 |5.74/80.80|
> | Spikformer  |AM-S  | 23.60/76.79 |5.74/82.90|
> | Spike-driven Transformer  |Baseline | 10.28/78.40 |2.57/80.00|
> | Spike-driven Transformer  |EMSP | 10.30/79.81|2.57/81.10|
> | Spike-driven Transformer  |AM-F  | 22.91/79.64 |5.74/81.50|
> | Spike-driven Transformer  |AM-S  | 22.91/78.89 |5.74/81.09|
> | Spikingformer  |Baseline | 9.32/79.21 |2.57/81.30|
> | Spikingformer  |EMSP | 9.39/79.81|2.58/81.95|
> | Spikingformer  |AM-F  | 23.59/79.06 |5.74/81.00|
> | Spikingformer  |AM-S  | 23.59/78.81 |5.74/82.50|
>
>
> > ***Weakness 5**: In Figure 4, do the four graphs in each sub-figure relate to four images from the same class?*
>
> **AW5**: We speculate that you are referring to Fig. 5, not Fig. 4. The background image is the same for each subplot. We will add this explanation to the manuscript.
>
> [R1] Noam Shazeer, et al. "Outrageously large neural networks: The sparsely-gated mixtureof-experts layer." International Conference on Learning Representations (2017).
>
> [R2] Yanqi Zhou, et al. "Mixture-of-experts with expert choice routing." Advances in Neural Information Processing Systems (2022).

---

> > ### Comment · Reviewer_dEFi · 2024-08-10
> >
> > Thank you for your response.
> >
> > To reiterate my previous comments, my primary concern is whether the proposed method is indispensable for enhancing the performance of Spiking Neural Networks (SNNs).  As confirmed by the authors, the paradigm is derived from previous works on MoE in ANNs [R1, R2], my perspective is unchanged: the significance of this paper hinges on “**verifying whether techniques developed for Artificial Neural Networks (ANNs) remain effective in the context of SNNs**”.  Therefore, the verification of the essentiality of the proposed method is crucial for the significance of this paper.
> >
> > For this reason, I had previously suggested that demonstrating an improvement over the state-of-the-art (SOTA) method [1] would be necessary. However, I find the authors' results unconvincing for the following reasons:
> > 1. **Model Selection:** The authors chose the smallest model variant, which achieves a performance of 74.1%, whereas the best performance reported in [1] is 80.0% with the largest model.
> > 2. **Experiment Setup:** For the selected model, the highest performance (74.1%) is achieved with 4 time steps. However, the authors used only 1 time step without providing a compelling rationale. This contrasts with their claim that their method is a "universal plug-and-play paradigm that can reliably improve performance". This is quite unacceptable to me. And I would decrease my rating due to the questionable results.
> >
> > Given the two-week time (including the author-review discussion period), I believe there is sufficient time for the authors to present more convincing results using the appropriate model that achieves the top performance of 80.0%.

---

> ### Author Response · Authors · 2024-08-13
> **Rebuttal by Authors**
>
> Dear Reviewer,
>
> Based on your suggestion, after overcoming significant challenges in resource allocation, we provide a performance report of using SEMM on the largest Meta-SDT-V2 model:
> ### Table R1-4
> | Model | ImageNet（T=1） | ImageNet (T=4)|
> | -------- |:--------:|:--------:|
> | Meta-SDT-V2-55M DT  | 78.00    | 79.70    |
> | Meta-SDT-V2-55M KD  | 79.10    | 80.00    |
> | Meta-SDT-V2-54M with SEMM DT  | __79.94__      | __81.80__    |
>
> Here, DT stands for Direct Training, while KD denotes Knowledge Distillation Assisted Training. SDT-V2-54M with SEMM DT shows an improvement of 2.10% over the direct training of the original model, and 1.80% higher than the knowledge distillation training of the original model.
>
> SEMM can still achieve stable performance improvements on the largest Meta-SDT-V2 model while maintaining a similar number of model parameters, demonstrating its generalizability and effectiveness.
>
> The logs, accuracy curves, specific code, and weights have been posted on an anonymous external link:
> https://huggingface.co/SDTV2SEMM/Rebuttal-SDTV2-SEMM-55M
>
> Best regards
>
> Authors

---

### Author Rebuttal · Authors · 2024-08-07

Dear ACs and Reviewers,

We sincerely appreciate the valuable time and feedback provided by each reviewer. We have responded to each of their comments individually.

In the rebuttal, we use

> ***Weakness/Question/Limitation**: ...*

to summarize the reviewers' comments. Responses start with '__AW, AQ, AL__'.

Best regards

Authors

---

### Decision · Program_Chairs · 2024-09-25

**Decision:**

Accept (poster)

**Comment:**

The reviewers found that this paper presents a novel, interesting idea in a clear way. There was less consensus on the merits of the empirical validation presented, but after following the discussion I am convinced that the authors provided sufficient evidence to make the paper a useful contribution to the conference.

From the discussion among the reviewers, one minor point emerged that the authors should take care to fix in the final version of the paper. The explanation given in Appendix B for EMSA's parameter count in Table 1 seems incorrect. This is easily checked by comparison with the code given as supplementary material (for instance in `Supplementary Material/ImageNet/models_spikformer_v1_moe.py`, lines 68-124). In appendix B,
1. the authors say that the the output layer needs $N^2$ parameters. This is incorrect: the output layer maps a mixture of the output of the experts, which lives in dimension $d$, to an output of dimension $D$; therefore it has $dD=D^2/m$ parameters.
2. the authors say that $K$, $V$, and $Q$ all need $dD=D^2/m$ parameters each. This is correct only for $K$ and $V$ (which are shared among all experts) but incorrect for $Q$. Each experts has an independent $Q$, each of which has $dD$ parameters, so the total number of parameters associated with $Q$ is $mdD=D^2$.

In practice, the two errors cancel each other out in the formula reported in Table 1, so the formula is correct, but the explanation in Appendix B should be fixed, and the main text should highlight the fact that $K$ and $V$ are shared among experts.